# Study of the Annealing Effect of Starch/Polyvinyl Alcohol Films Crosslinked with Glutaraldehyde

**DOI:** 10.3390/gels7040249

**Published:** 2021-12-03

**Authors:** Edgar Franco, Rosmery Dussán, Diana Paola Navia, Maribel Amú

**Affiliations:** 1Grupo de Investigación Arquitectura, Urbanismo y Estética, Facultad de Arquitectura, Arte y Diseño, Universidad de San Buenaventura, Cali 760031, Colombia; director.disenovestuario@usbcali.edu.co; 2Grupo de Investigación Biotecnología, Facultad de Ingeniería, Universidad de San Buenaventura, Cali 760031, Colombia; dpnavia@usbcali.edu.co (D.P.N.); mamu@usbcali.edu.co (M.A.)

**Keywords:** polyvinyl alcohol, cassava starch, annealing, crosslinked, response surface analysis

## Abstract

Films were fabricated using a mixture of polyvinyl alcohol (PVA)/cassava starch and incorporated citric acid in a concentration range between 5% and 40%. The films were annealed through thermal treatment in a temperature range between 30 °C and 90 °C with 0.3% glutaraldehyde incorporated as the crosslinking agent. This study presents the results of an experimental design analyzed using the response surface methodology. The multiple regression analysis allowed us to obtain the second-order models, which relate the annealing factors and citric acid concentration to Maximum Tensile Strength (*MTS*), Young’s Modulus (*YM*), and the Maximum Elongation at Break (*MEB*). The optimization and validation of the obtained model were carried out with error values below 10.08% for all the response variables, indicating that the response surface methodology and optimization were correct. Finally, as a complementary analysis, the differential scanning calorimetry (DSC) and Fourier-transform infrared spectroscopy (FTIR) tests were carried out, which revealed a higher packaging of the heat-treated films and verified their crosslinking.

## 1. Introduction

The current trend of replacing synthetic polymers with natural polymers obtained from renewable resources has increased both the implementation of studies related to natural polymers of plant origin, such as cassava starch, and the search for improvement of its specific properties for application mainly in packaging and crating. Recent research has studied the mixture of starch and poly(vinyl alcohol) (PVA), because PVA is a biodegradable, water-soluble, non-toxic, synthetic polymer with hydroxyl groups, it is similar to starch and plasticizers such as glycerol, sorbitol, and citric acid [1,2]. These studies evaluated the starch/PVA blend with different additives and the modification of process conditions. Films obtained were mechanically resistant, stable, and non-toxic for possible use in food packaging. Tanwar et al. [3], discussing this application, stated the results of the development of a biocomposite obtained by mixing PVA/starch with sorbitol, sepiolite clay, and coconut shell extract. Films showed antioxidant properties and an improvement in their mechanical properties, demonstrating their potential use in food packaging [3]. In recent studies, composite films based on PVA/starch, zinc oxide nanoparticles, and phytochemicals were prepared using a casting method. Films showed a higher water barrier and better mechanical and antimicrobial properties, which led to the conclusion that they have potential use in food packaging [4]. In the results presented by Priya et al. [5], films were prepared through the casting method using PVA/corn starch mixtures, which were plasticized with citric acid, crosslinked with glutaraldehyde, and reinforced with cellulose fibers. These composite materials were characterized mechanically through tensile testing, swelling testing, solubility testing, FTIR, and DCS. It was found that the films achieved good thermal, mechanical, and antimicrobial activity, which gave them a high potential for use in food packaging.

This study applies a similar design to the one discussed above, but it performed an annealing heat treatment, which allows control of the final mechanical properties. Therefore, an experimental design analyzed through the response surface methodology was proposed. Films made from the PVA/cassava starch polymer mixture, plasticized with citric acid in a concentration range between 5% and 40%. Citric acid was chosen because it is an economical and non-toxic chemical; it makes the films achieve higher mechanical strength compared to plasticizers such as glycerol [6]. The addition of glutaraldehyde was constant at 0.3 wt%, seeking higher film integrity under humid conditions, and improved thermal and mechanical properties [7]. This aldehyde can crosslink PVA through the acetalization reaction [8], which Priya and others also reported on the crosslinking of starch/PVA mixtures [5]. Due to the fact that PVA is affected by temperature rises [9,10], 30-min annealing heat treatments were performed on the prepared films in a temperature range between 30 °C and 90 °C.

The objective of this study was to obtain an optimized and validated mathematical model to predict the citric acid concentration and annealing temperature necessary to obtain films with mechanical properties that allow their application as packaging. Therefore, the maximum resistance, the modulus of elasticity, and maximum deformation obtained from the tensile test were evaluated as response variables in a central composite design. Finally, as a complementary and explanatory evaluation, FTIR and DSC tests were carried out.

## 2. Results and Discussion

### 2.1. Response Surface Analysis (RSM)

The RSM was used to analyze the effect of the annealing and the concentration of citric acid on the Maximum Tensile Strength (*MTS*), Young’s Modulus (*YM*), and Maximum Elongation at Break (*MEB*) mechanical properties. Table 1 shows the results obtained from the central composite design (CCD).

The models proposed for each response variable are correlated with the assumption about errors. Errors were normally distributed, with zero mean and σ^2^ variance, at significance levels of up to 5%. Table 2 shows the analysis of variance.

The value of the regression sum of squares (SSR) is higher than the value of the error sums of squares (SSE) for all response variables. This indicates that there is a higher contribution of the regression in contrast to the error one. This was verified by the values of the coefficient of determination. The *p*-value of the regression was significant for all models with values, i.e., below 0.009.

The hypothesis test with the value of the lack of fit was used to verify the model fit. The null hypothesis indicates that the predicted model is adequate, and the alternative one indicates the opposite. The test result did not show evidence of lack of fit of models, with significance levels greater than 20.2%.

The coefficient of determination, R^2^, was used to demonstrate the fit of the quadratic models of each dependent variable. R^2^ values were higher than 0.84, indicating a good fit. In the case of the *MTS*, 98.27% represents the variation explained by the *X*(*_f_*), its squares, and its interactions. Similarly, 98.98%, and 84.61% for *YM* and *MEB*, respectively.

Multiple regression analysis was used to obtain the second-order models, which relate the annealing factors and citric acid concentration with *MTS*, *YM*, and *MEB*. The model equations for the response variables are:(1)$$MTS=58.42-1.591X_{1}-0.588X_{2}+0.02083X_{1}X_{1}+0.00861X_{2}X_{2}-0.00562X_{1}X_{2}$$
(2)$$YM=2354-101.9X_{1}-21.24X_{2}+1.579X_{1}X_{1}+0.3629X_{2}X_{2}-0.336X_{1}X_{2}$$
(3)$$MEB=-428+10.8X_{1}+14.11X_{2}+0.274X_{1}X_{1}-0.1016X_{2}X_{2}-0.219X_{1}X_{2}$$

Table 3 shows the effects of the regression coefficients for the dependent variables. At significance levels below 5%, the *p*-values marked with (*) were significant. *MTS* and *MEB* showed non-significant effects that were not rejected, because they were meaningful for *YM*.

### 2.2. Response Surface

The response surfaces (Figure 1, Figure 2 and Figure 3) were obtained from Equations (1)–(3).

### 2.3. Optimization and Validation of Models

The Maximum Tensile Strength, Young’s Modulus, and Maximum Elongation at Break response variables were optimized. The target values for each independent variable shown in Table 4 obey values in the range of the properties of plastics, such as polyethylene, used in packaging and crating applications, which, for the effects of the mechanical properties of the analyzed material, are considered sufficient for this application [11,12,13].

Table 4 shows the optimization conditions for each independent variable. A minimum or maximum value and a target value were established for each variable.

Figure 4 shows the solution of the optimized model for each *MTS*, *YM*, and *MEB* independent variable. The model presents a high prediction percentage (87.9%). The predicted values for *MTS*, *YM*, and *MEB* were 31.09 MPa, 1003.84 MPa, and 62.72%, respectively, with high predictability (d > 0.7194) for citric acid (12%) and annealing (50 °C). These results are similar to that reported by Tanwar et al. [3] concerning *MTS* and four times higher than that study’s *MEB* results. However, that study used corn starch, sorbitol as a plasticizer additive, and no annealing [3].

### 2.4. Validation of the Optimal Conditions

The fit of the models in Equations (1)–(3) was verified through the validation of the optimized response variables. The films made under the process conditions resulted in the optimization used to compare the experimental and predicted values shown in Table 5. Error values were lower than 10.08% for all response variables, indicating that the response surface methodology and optimization were correct. Therefore, it means that the results have adequate statistical support.

Results concerning the behavior of the material indicate that the increase in the temperature of heat treatment showed a tendency to decrease *MEB* and increase *MTS* and *YM*. Finally, an increase in the concentration of the variable CA effectively fulfils the functions of a plasticizer additive, since it reduces the *MTS* and the *YM* but increases the *MEB* [5]. The annealing performed in the temperature range between 30 °C to 90 °C led to a higher magnitude of dehydration of the films as the temperature was higher. This treatment led to a mass reduction from 1.35 g (±0.16) to 1.29 g (±0.15) and an average film thickness of 0.13 mm (±0.03), obtained after annealing. This behavior occurred, because annealing increases the crystallinity of the films by removing residual water and forming new hydrogen bonds between the macromolecules [9].

### 2.5. DSC Thermal Analysis

Figure 5 shows the DSC thermograms. The PVA curve illustrates the characteristic semicrystalline behavior of this polymer with a glass transition temperature (T_g_) of 76.79 °C, a melting point (T_m_) of 168.83 °C, and an enthalpy of fusion (ΔH_m_) of 26.25 J/g, while, for cassava starch, no exothermic peak was observed, which indicates an amorphous state. Amorphous behavior was observed in the mixture without heat-treated starch/PVA. This occurred, because the two polymers were dissolved and then mixed during their preparation, which introduced amorphousness into the system.

Comparing the enthalpy of the (starch/PVA) sample with the (starch/PVA HT) sample shows an increase from 0.060 J/g to 6.74 J/g, respectively. The appearance of this slight peak shows a rearrangement and increase in crystallinity due to the annealing, which leads to the hydroxyl groups in the mixture generating hydrogen bonds between PVA, starch, and citric acid [14]. The small exothermic peak of starch/PVA HT also verifies the crosslinking of both citric acid–starch and PVA-glutaraldehyde [15].

### 2.6. FTIR Spectroscopy

FTIR spectra results (Figure 6) show a broad peak at 3431.39 cm^−1^ due to bound OH groups. The absorption peaks at 2922.31, 1431.01, and 1021.93 cm^−1^ occurred due to the stretching of CH_2_, CC, and CO, respectively [5].

Figure 6 shows the comparison between the formed films, both heat-treated (starch/PVA HT) and not treated (starch/PVA), containing starch and PVA. A peak in the 1742 cm^−1^ band was identified, this was attributed to the formation of carboxyl and carbonyl ester, which confirms the chemical bond formed between citric acid and starch [6]. This also indicates a large number of vibrational modes of interactions between starch and PVA [16,17].

## 3. Conclusions

The multiple regression analysis allowed us to obtain the second-order models, which relate the annealing factors and citric acid concentrations with *MTS*, *YM*, and *MEB*, with a high prediction percentage (87.9%). For the predicted values of the dependent *MTS*, *YM*, and *MEB* (31.09 MPa, 1003.84 MPa, and 62.72%, respectively) the optimal combination of the citric acid and annealing concentration variables was 12% and 50 °C, respectively, with a significance of *p* < 0.05. This mathematical model allows users to predict the mechanical properties of films, which can be used as packaging, based on user requirements. Finally, the DSC analysis revealed a higher packaging of the heat-treated and crosslinked films, both of CA–CS and glutaraldehyde–PVA. This latter crosslinking was also confirmed through the FTIR test, in addition to having been developed before heat treatment.

## 4. Materials and Methods

Cassava starch (CS) from Tecnas S.A, Colombia brand was used; poly(vinyl alcohol) (PVA) hydrolyzed 98–99%, Mw = (146,000–186,000) from Sigma-Aldrich, St. Louis, MO, USA, with a high degree of hydrolysis. In addition, glutaraldehyde (GLU) was used as a crosslinking agent, and commercial citric acid (CA) was used as a plasticizer additive.

### 4.1. Preparation of Films Using the Casting Method

Two aqueous polymer solutions were prepared. The first PVA solution was at 10% weight, heating up to 90 °C with magnetic stirring, and the second cassava starch solution was at 10% weight. In the latter, citric acid was added as a plasticizer additive in a ratio (CA/(CA + CS + PVA)) × 100 and in a concentration range of 5% to 40% and heated up to 80 °C with magnetic stirring. Subsequently, the two solutions were mixed in equal amounts, stirring for 5 min, then glutaraldehyde was added in a percentage of 0.3 wt%, according to the total amount of polymer (CS + PVA) [5]. Finally, this solution was stirred for an additional 15 min and then 10 g of it was poured inside plastic Petri dishes of 90 mm diameter. Subsequently, the films were dried for five days at room temperature (25 °C) and stabilized at a relative humidity of 43.85% (±3.26). Then, the heat treatments were carried out, as described in the design of the experiment defined below.

### 4.2. Experiments Design

A central composite design (CCD) with two independent variables was used to measure the effectiveness of the mechanical properties: *X*_1_: 5–40% citric acid and *X*_2_: annealing 30–90 °C (Table 6). The experimental data were obtained from 13 experimental runs. The data were fitted to the second-order polynomial model:(4)$$Y=\beta_{0}+\beta_{1}X_{1}+\beta_{2}X_{2}+\ldots +\beta_{3}X_{1}^{2}+\beta_{4}X_{2}^{2}+\ldots +\beta_{5}X_{1}X_{2}+\ldots +\varepsilon$$
where *Y* is the response variable: Maximum Tensile Strength (*MTS*) (MPa), Young’s Modulus (*YM*) (MPa), and Maximum Elongation at Break (*MEB*) (%); *βi* represents the model coefficients for each predictive variable; *X_(f)_* represents independent variables (*f* = 1 … 4); *X_(f)_^2^* are the quadratic effects of each predictive variable; and *X_(f)_X_(f)_* is the interaction effect between the variables and ε is the experimental error.

### 4.3. Statistical Analysis and Optimization

The mathematical model, the analysis of variance, and the optimization were determined using Minitab statistical software (17th version, State College, Pennsylvania, EE. UU) and the response surface methodology (RSM). The lack of fit and the coefficient of determination (R^2^) were the criteria used to establish the fit of the second-order model with a confidence level of 95%. Additional experiments were carried out to verify the validity of the statistical results.

The desirability function (D: global; d: individual) was used for optimization. It converts the functions to a scale between 0 and 1, combining them with the geometric means and optimizing the general metric means [18].

### 4.4. Films Characterization

#### 4.4.1. Tensile Test

Maximum resistance, modulus of elasticity, and maximum deformation were obtained through the tensile test, following the ASTM D 882-02 standard, tested at a speed of 20 mm/min.

#### 4.4.2. DSC Calorimetry and FTIR Spectroscopy Tests

These tests were carried out on the validated films to evaluate the effect of the heat treatment. Therefore, samples with and without heat treatment were taken. A Differential Scanning Calorimetry (DSC) test was conducted at a heating rate of 20 °C/min, which allowed the thermal analysis of the prepared and treated films in a temperature range of 50 °C to 250 °C. In this test, the analyzed thermograms were taken during a second scan. The FTIR test was performed with the Nicolet 6700 FT-IR equipment.

## Figures and Tables

**Figure 1 gels-07-00249-f001:**
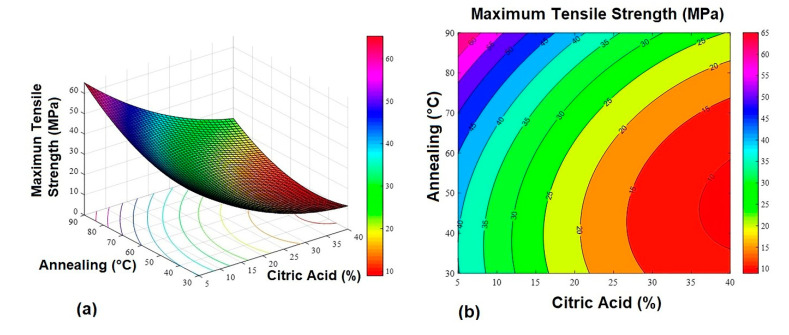
Maximum tensile strength response surface graphs. (**a**) 3D response surface graph and (**b**) contour plot.

**Figure 2 gels-07-00249-f002:**
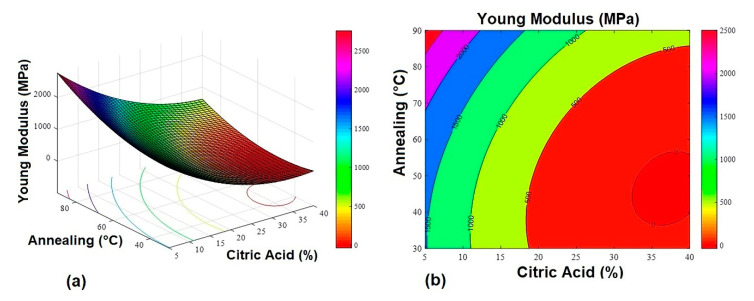
Young’s modulus response surface graphs. (**a**) 3D response surface graph and (**b**) contour plot.

**Figure 3 gels-07-00249-f003:**
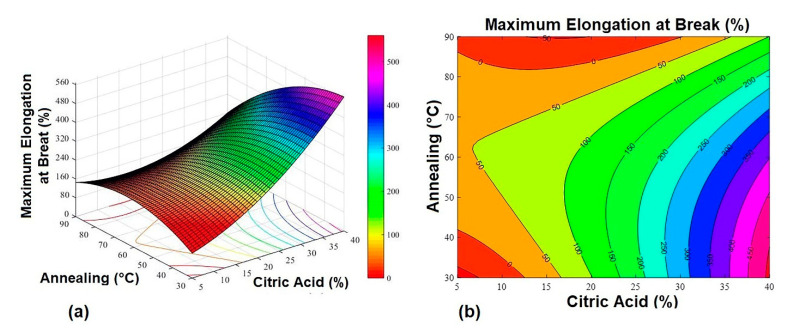
Maximum elongation at break response surface graphs. (**a**) 3D response surface graph and (**b**) contour plot.

**Figure 4 gels-07-00249-f004:**
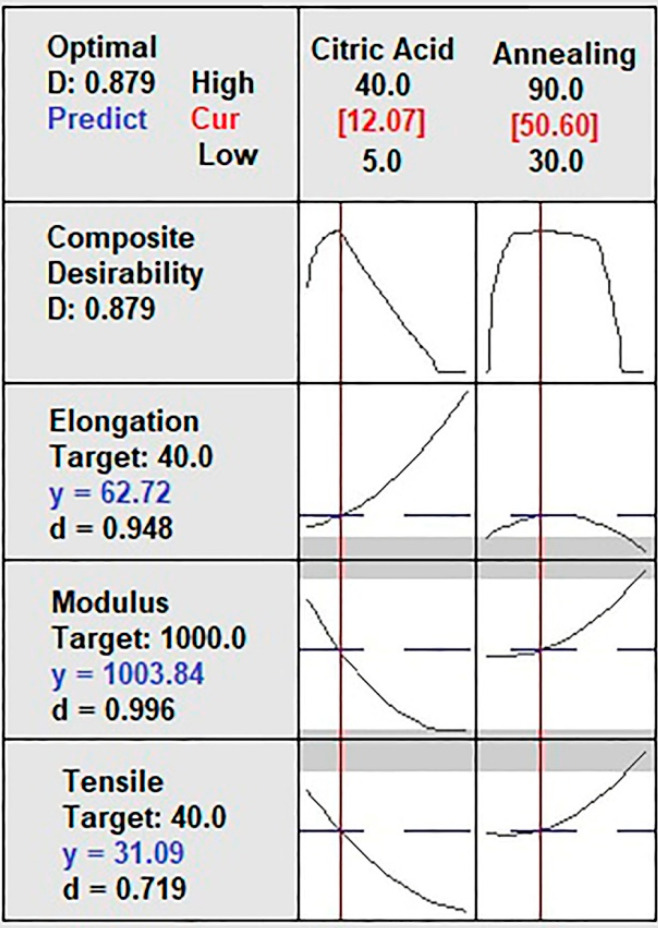
Optimized model solution for mechanical properties of the film.

**Figure 5 gels-07-00249-f005:**
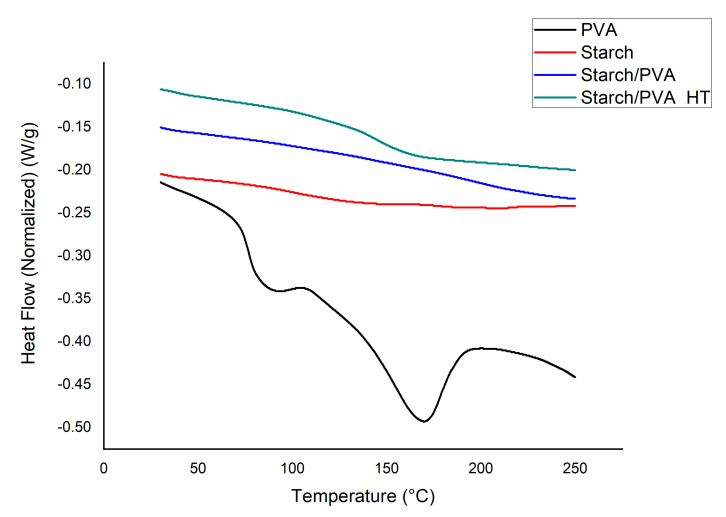
DSC thermograms of PVA, starch/PVA and starch/PVA HT.

**Figure 6 gels-07-00249-f006:**
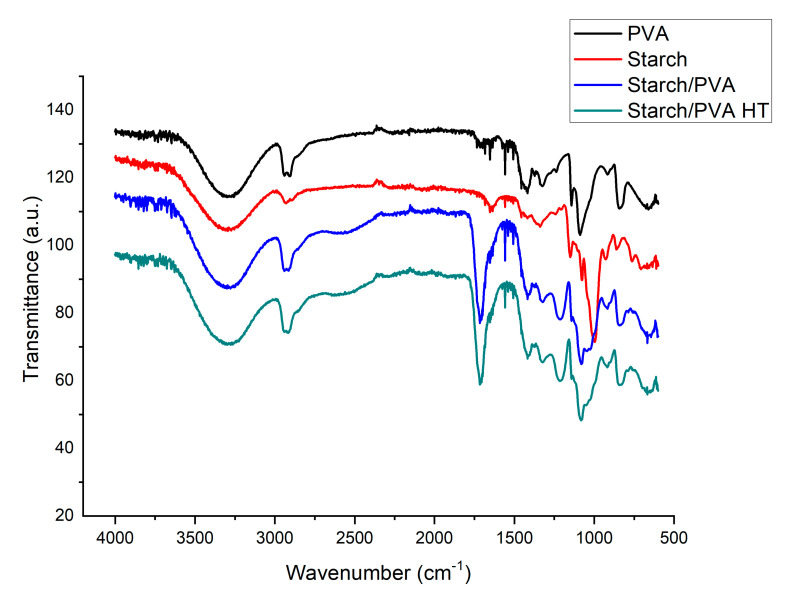
FTIR spectra of PVA, starch, starch/PVA and starch/PVA HT.

**Table 1 gels-07-00249-t001:** Results of mechanical properties in central composite design.

Run	Independent Variables		Maximum Tensile Strength (MPa)	Young’s Modulus (MPa)	Maximum Elongation at Break (%)
	Citric Acid (%) *X*_1_	Annealing (°C) *X*_2_			
1	22.50	90.00	39.40	1237.79	2.62
2	22.50	60.00	21.61	437.05	126.88
3	22.50	60.00	20.96	352.69	254.24
4	22.50	60.00	23.03	541.89	81.32
5	34.87	38.78	12.83	4.39	330.81
6	22.50	60.00	18.69	363.62	142.10
7	40.00	60.00	8.28	1.34	477.50
8	10.12	38.78	31.46	1008.85	40.66
9	34.87	81.21	22.24	451.44	63.12
10	22.50	60.00	22.13	504.90	91.96
11	5.00	60.00	47.09	1899.20	22.79
12	10.12	81.21	46.77	1808.60	2.58
13	22.50	30.00	18.70	349.10	147.18

**Table 2 gels-07-00249-t002:** Analysis of variance of mechanical properties in central composite design.

Source	Maximum Tensile Strength (MPa)	Young’s Modulus (MPa)	Maximum Elongation at Break (%)
SSR	1730	4,526,291	199,931
*p*-value	0.0000	0.0000	0.0090
SSE	30.4900	46794	36,369
Lack of fit (*p*-value)	0.2020	0.5190	0.4020
R^2^	0.9827	0.9898	0.8461

**Table 3 gels-07-00249-t003:** Analysis of variance for regression coefficients of mechanical properties.

Variable	*p*	Maximum Tensile Strength (MPa)		Young’s Modulus (MPa)		Maximum Elongation at Break (%)	
		E	*p*-Value	E	*p*-Value	E	*p*-Value
	*β* _0_	58.4200	<0.0001 *	2354.0000	<0.0001 *	−428.0000	0.0030 *
Linear							
*X* _1_	*β* _1_	−1.5910	<0.0001 *	−101.9000	<0.0001 *	10.8000	0.0020 *
*X* _2_	*β* _2_	−0.5880	<0.0001 *	−21.2400	<0.0001 *	14.1100	0.0410 *
Quadratic							
*X* _1_	$\beta_{1}^{2}$	0.0208	0.0050 *	1.5790	<0.0001 *	0.2740	0.1690
*X* _2_	$\beta_{2}^{2}$	0.0086	0.0020 *	0.3629	0.0010 *	−0.1016	0.1380
Interaction							
*X* _1_ *X* _2_	*β* _12_	−0.0056	0.2000	−0.3360	0.0680	−0.2190	0.1550

* Significantly different at *p* < 0.05; p: Parameters; E: Estimated coefficient value.

**Table 4 gels-07-00249-t004:** Optimization conditions for mechanical properties of the film.

Response Variable	Minimum	Maximum	Goal	Objective
Maximum Tensile Strength (MPa)	8.28	47.09	Objective	40.00
Young Modulus (MPa)	1.34	1899.20	Objective	1000.00
Maximum Elongation at Break (%)	2.63	477.57	Objective	40.00

**Table 5 gels-07-00249-t005:** Experimental validation of mechanical properties at the optimal conditions for the films.

Response Variable	Predicted Value	Experimental Value *	Error ** (%)
Maximum Tensile Strength (MPa)	31.09	33.11 ± 1.15	7.96
Young’s Modulus (MPa)	1003.84	1007.12 ± 13.76	0.33
Maximum Elongation at Break (%)	62.72	56.98 ± 2.23	10.08

* Experimental values were obtained under the following conditions: Citric acid (12%)—Annealing 50 °C. ** Error = [(experimental value−predicted value)/experimental value)] × 100.

**Table 6 gels-07-00249-t006:** Code variables in experimental design for film preparation.

Code Variables	Citric Acid	Annealing
	(%)	(°C)
−1	5.00	30.00
0	22.50	60.00
1	40.00	90.00

## Data Availability

Not applicable.

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
