# Peer review of "Study of the Annealing Effect of Starch/Polyvinyl Alcohol Films Crosslinked with Glutaraldehyde"

_gels, 2021, doi:10.3390/gels7040249_

Round 1

Reviewer 1 Report

The authors report their studies on annealed crosslinked starch/ PVA films. The paper covers crosslinked polymeric composite films, which may represent gels in dry state, therefore can be considered for publication in Gels. However, the paper may benefit a greater audience if published in journals such as Materials or Polymers.

The paper can be improved by attending to the following:

  1. The introduction should be expanded to capture recent studies on PVA/ starch films.
  2. The introduction should clearly state what research gap the current study aims to fill? What is novel about the current study?
  3. Section 2.1: Please specify if the 0.3% glutaraldehyde is based on weight?
  4. Section 2: please specify the exact degree of hydrolysis for PVA.
  5. Section 2.1: Please clarify if the solvent was evaporated to form films, which were heat treated? The text currently depicts that the solution was heat treated.
  6. Please specify if second DSC scans were used in the study? Usually a first scan is done to eliminate moisture in polymer films.

Author Response

Response to Reviewer 1 Comments

The paper can be improved by attending to the following:

Point 1: The introduction should be expanded to capture recent studies on PVA/ starch films

 Response 1: Three publications with recent studies on PVA/starch films were included between lines 28 to 42 in the introduction.

Point 2: The introduction should clearly state what research gap the current study aims to fill? What is novel about the current study?

Response 2: In the introduction, the research gap the current study aims to fill was included in lines 63 to 65.

Point 3: Section 2.1: Please specify if the 0.3% glutaraldehyde is based on weight?

Response 3: The 0.3% is based on the weight of the glutaraldehyde. It has been specified between lines 56 and 82.

Point 4: Section 2: please specify the exact degree of hydrolysis for PVA

 Response 4: The exact degree of hydrolysis for PVA is 98-99%. It was specified between lines 71 to 72.

Point 5: Section 2.1: Please clarify if the solvent was evaporated to form films, which were heat-treated? The text currently depicts that the solution was heat treated.

 Response 5: The solvent was evaporated to form films. It was specified between lines 248 to 251.

Point 6: Please specify if second DSC scans were used in the study? Usually, a first scan is done to eliminate moisture in polymer films.

 Response 6: The thermograms correspond to the second DSC scan. This was clarified on line 128.

Please see attached manuscript with the changes

Reviewer 2 Report

The study's objective is not presented, the importance of the polymers used, nor is the reason why the crosslinking agent is added if there are already studies of materials based on PVA-starch without crosslinking.

It is necessary to do a language revision of the entire document.

Line 11: Change uppercase to lowercase in word Cassava

Line 15: Change uppercase to lowercase in words annealing, citric and acid

Line 29-36: Present the relevant information of this study more consistently since it is given more importance than the study developed by the manuscript's authors

Line 37: Change "this article" by "this study". Improve the wording of the study's objective presented in this manuscript, if it is possible to add the hypothesis of the study. Add previous studies by other authors. What is the advantage of using citric acid as a plasticizer instead of conventional plasticizers for this type of material such as glycerol, sorbitol, or PEG?

Line 60: Why was that amount of crosslinking added? Preliminary studies of the authors or any published document? The process used to carry out the crosslinking is not explained.

Line 61: What was the drying process of the filmogenic solutions poured into the Petri dishes, temperature, time of obtaining or drying, and equipment used

Line 62: Mention the material and size of the Petri dishes; it is essential to add that information

Line 117: Standardize the number of decimal places in tables because tables have 2, 3, and up to 4 decimal places.

Line 247: Avoid expressions like “The following figure”, mention it as Figure 2 or the one that corresponds; review it throughout the document.

Line 270: Change the word “greater”

Author Response

Response to Reviewer 2 Comments

Point 1: The study's objective is not presented, the importance of the polymers used, nor is the reason why the crosslinking agent is added if there are already studies of materials based on PVA-starch without crosslinking.

 Response 1: The study's aim was incorporated between lines 63 to 65.

The importance of the polymers used was included between lines 27 to 30.

The importance of the crosslinking agent used was included between lines 56 to 59.

Point 2: It is necessary to do a language revision of the entire document.

 Response 2: A review and correction of the language throughout the document were carried out.

Point 3: Line 11: Change uppercase to lowercase in word Cassava

Response 3: The word cassava has been changed from uppercase to lowercase throughout the document.

Point 4: Line 15: Change uppercase to lowercase in words annealing, citric and acid

Response 4: The words: annealing, citric and acid, were been changed from uppercase to lowercase throughout the document.

Point 5: Line 29-36: Present the relevant information of this study more consistently since it is given more importance than the study developed by the manuscript's authors

Response 5: In the introduction, the importance of this study was expanded upon in lines 32 to 45.

Point 6: Line 37: Change "this article" by "this study". Improve the wording of the study's objective presented in this manuscript, if it is possible to add the hypothesis of the study. Add previous studies by other authors. What is the advantage of using citric acid as a plasticizer instead of conventional plasticizers for this type of material such as glycerol, sorbitol, or PEG?

Response 6:

In line 49 the word "article" was changed to the word "study".

In the introduction, the wording of the study's aim was improved in lines 63 to 65.

Three publications with recent studies on PVA/starch films were included between lines 28 to 42 in the introduction.

Between lines 53 to 59, the use of citric acid in the preparation of the films was justified.

Point 7: Line 60: Why was that amount of crosslinking added? Preliminary studies of the authors or any published document? The process used to carry out the crosslinking is not explained.

Response 7: Between lines 56 to 59, a study was cited that justifies the amount of crosslinking added.

Point 8: Line 61: What was the drying process of the filmogenic solutions poured into the Petri dishes, temperature, time of obtaining or drying, and equipment used

Response 8: Line 85 describes the time, temperature, and equipment used in drying the filmogenic solutions.

Point 9: Line 62: Mention the material and size of the Petri dishes; it is essential to add that information

Response 9: Line 84 describes the material and size of the Petri dishes

Point 10: Line 117: Standardize the number of decimal places in tables because tables have 2, 3, and up to 4 decimal places.

Response 10: The number of decimal places in tables was standardized. Tables 3 and 4 require high precision in the p-value, determination, and regression coefficients. In tables 3 and 4, 4 decimal places were standardized, and in the other tables, 2 decimal places were standardized. 

Point 11: Line 247: Avoid expressions like “The following figure”, mention it as Figure 2 or the one that corresponds; review it throughout the document.

Response 11: Expression “The following figure” was eliminated in line 284. This expression was reviewed throughout the document.

Point 12: Line 270: Change the word “greater”

Response 12: In line 260 the word "greater" was changed to the word "better".

Please see attached manuscript with the changes

Reviewer 3 Report

  1. Introduction should be modified and more related references and previous related works should be discussed and compared considering the aim of this work.
  2. The objectives and novelty of the work should be clearly explained at the end of the Introduction.
  3. The exact method of the annealing should be more clearly explained in the Experimental. Only the temperatures have been mentioned.
  4. The titles of the figures and tables are not complete. they should be more complete and informative.
  5. Citric Acid (CA) and not AC.
  6. More references can be cited.

Author Response

Response to Reviewer 3 Comments

Point 1: Introduction should be modified and more related references and previous related works should be discussed and compared considering the aim of this work.

 Response 1:

Four references with recent studies on PVA/starch films were included between lines 27 to 40 in the introduction.

Three references with recent studies on citric acid as a crosslinking agent used were included between lines 53 to 59 in the introduction.

Point 2: The objectives and novelty of the work should be clearly explained at the end of the Introduction

Response 2:

At the end of the introduction, the wording of the study's aim was improved in lines 63 to 65.

In the introduction, the research gap the current study aims to fill was included in lines 63 to 69.

Point 3: The exact method of the annealing should be more clearly explained in the Experimental. Only the temperatures have been mentioned.

Response 3: The description of the annealing method was improved in lines 84 and 85.

Point 4: The titles of the figures and tables are not complete. they should be more complete and informative

Response 4: The titles of the figures y tables were complemented

Point 5: Citric Acid (CA) and not AC.

Response 5: The acronym AC was changed to CA throughout the document.

Point 6: More references can be cited

Response 6: 7 new references were included in the introduction.

Please see attached manuscript with the changes

Round 2

Reviewer 1 Report

The authors have attended to the reviewers comments and have improved the manuscript significantly. The manuscript may now be published, after minor text editing.

Reviewer 3 Report

Most of the comments have been answered by the authors and the manuscript can be accepted.